# The Real Need for Regenerative Medicine in the Future of Congenital Heart Disease Treatment

**DOI:** 10.3390/biomedicines9050478

**Published:** 2021-04-27

**Authors:** Yuichi Matsuzaki, Matthew G. Wiet, Brian A. Boe, Toshiharu Shinoka

**Affiliations:** 1Center for Regenerative Medicine, The Abigail Wexner Research Institute at Nationwide Children’s Hospital, 700 Children’s Drive, T2294, Columbus, OH 43205, USA; Yuichi.Masuzaki@nationwidechildrens.org (Y.M.); Matthew.Wiet@nationwidechildrens.org (M.G.W.); 2Department of Cardiology, The Heart Center, Nationwide Children’s Hospital, 700 Children’s Drive, T2294, Columbus, OH 43205, USA; Brian.Boe@nationwidechildrens.org; 3Department of Cardiothoracic Surgery, The Heart Center, Nationwide Children’s Hospital, 700 Children’s Drive, T2294, Columbus, OH 43205, USA

**Keywords:** tissue engineering, large animal model, pulmonary valve, aortic patch, pulmonary vein

## Abstract

Bioabsorbable materials made from polymeric compounds have been used in many fields of regenerative medicine to promote tissue regeneration. These materials replace autologous tissue and, due to their growth potential, make excellent substitutes for cardiovascular applications in the treatment of congenital heart disease. However, there remains a sizable gap between their theoretical advantages and actual clinical application within pediatric cardiovascular surgery. This review will focus on four areas of regenerative medicine in which bioabsorbable materials have the potential to alleviate the burden where current treatment options have been unable to within the field of pediatric cardiovascular surgery. These four areas include tissue-engineered pulmonary valves, tissue-engineered patches, regenerative medicine options for treatment of pulmonary vein stenosis and tissue-engineered vascular grafts. We will discuss the research and development of biocompatible materials reported to date, the evaluation of materials in vitro, and the results of studies that have progressed to clinical trials.

## 1. Introduction

Congenital heart disease (CHD) accounts for approximately one-third of all congenital anomalies worldwide affecting approximately 40,000 newborns in the United States each year [1,2]. Despite advances in management, CHD remains a leading cause of death in newborns and can lead to lifelong morbidity in survivors [1]. A significant portion of cardiac surgery morbidity and mortality is attributed to the synthetic conduits and patches frequently used for cardiac repair. Made from materials such as polytetrafluoroethylene (PTFE or Gore-Tex), these artificial materials are susceptible to thrombogenicity, stenosis, iatrogenic calcification, immune rejection, and infection [3,4]. Grafts constructed from these materials also lack growth potential, contributing to one of the greatest causes of morbidity in pediatric patients—somatic overgrowth, or the process by which patients outgrow their grafts. Alternative material choices have largely revolved around allografts, xenografts, and autologous tissues such as pericardium or saphenous vein; however, each of these have varying degrees of complications and still do not address the need for growth potential [5].

Regenerative medicine provides a potential solution to the use of these various materials, allowing for the added benefit of avoiding issues such as immune rejection, somatic overgrowth, infection, and calcification [6,7]. This is made possible through the strategy of implanting a bioabsorbable scaffold that degrades over time which is replaced with autologous vascular tissue that can repair, reform, and even grow with the patient [8]. Many groups have sought to create different surgical materials [9,10], but have been met with numerous obstacles that must be addressed in order for these materials to be acceptable for clinical practice. For this reason, very few cardiovascular materials utilizing regenerative medicine principles have made it to the clinical trial stage [11]. It is our sense that there is an unintended disconnect between the materials being studied by materials science experts/engineers and those found most useful by surgeons implanting them. In this article, we will discuss the real needs of regenerative medicine from the standpoint of pediatric cardiovascular surgery, focusing on the current status of regenerative engineering valves, regenerative engineering patches, and regenerative medicine for pulmonary vein stenosis using bioabsorbable materials [12,13].

## 2. Tissue-Engineered Pulmonary Valve

The pulmonary valve (PV) is the most commonly affected valve in CHD. Separating the right ventricle (RV) and pulmonary trunk, the PV is a tri-leaflet structure leading to the lower pressure pulmonary system. It serves to facilitate forward blood flow during systole and prevent retrograde blood flow during diastole. The regulation of flow is a function of its geometric shape, cellular composition, and extracellular matrix (ECM). Geometrically, leaflet alignment via the commissures and annulus determines appropriate coaptation of the PV and thus prevention of regurgitation. Valve leaflets are complex structures composed of various cells types that create and maintain an intricate ECM based on environmental stimuli [14]. As such, conditions (congenital or iatrogenic) affecting these valvular components place the patient at increased risk of significant morbidity and mortality, often times necessitating PV replacement (PVR) [15,16,17,18,19].

Isolated PVR in a CHD patient is much less common than a PVR that is a result of sequelae from right ventricular outflow tract (RVOT) modification during initial repair of the CHD lesion [20]. In fact, the majority of two-chamber endocardial repairs for CHD involve repair of a dysfunctional RVOT using a valved conduit or bioprosthetic pulmonary valve (Table 1). Some of the most commonly used conduits are decellularized homografts, conduits with valves made of e-PTFE, or the glutaraldehyde-fixed, bovine-jugular vein-derived Contegra^®^ valved conduit. Long-term complications related to modification of the RVOT during endocardial repair are the leading cause of catheterization and/or reoperation in the distant postoperative period for these patients [21]. These complications include potential for infection due to non-autologous tissue, deterioration of valve function from calcification, associated pulmonary valve stenosis, and right heart failure due to pulmonary valve regurgitation. Ideally, the goal of the initial RVOT reconstruction and conduit placement in CHD patients would be avoidance of repeat surgical or transcatheterization intervention. However, this is limited by current treatment options and materials. To prevent right heart failure from PV regurgitation, it is important to perform either surgical or transcatheter PVR before the right ventricular function is irreversibly compromised.

PVR traditionally involves either surgical or transcatheter approaches using an array of different mechanical, allo-/xenogenic, or synthetic valve substitutes each with their own varying levels of success. For surgical PVR, bioprosthetic (mainly decellularized) homograft valves have been a widely utilized valve choice due to their decreased thrombogenicity and the lack of required lifelong anticoagulation associated with mechanical valves. However, their apparent short lifespan and tendency to calcify/degenerate has been a known issue for many years, requiring multiple reinterventions throughout a pediatric patient’s lifetime [23,24]. The more recently accepted approach for PVR emerged in 2000 as Bonhoeffer performed the first transcatheter valve replacement. A native, valved bovine jugular vein sewn onto a stent was implanted into the conduit of a 12-year-old boy with repaired pulmonary atresia and a ventricular septal defect [25]. Since then, the field of transcatheter PVR has grown substantially with the two most common transcatheter PV being the Melody^®^ Transcatheter Pulmonary Valve (Medtronic, Minneapolis, MN, USA) [26,27] and the Sapien Transcatheter Heart Valve (Edwards Lifesciences, Irvine, CA, USA) [28]. The Melody^®^ valve is constructed from a bovine jugular vein, whereas the Sapien valve is constructed with bovine pericardium. Both valves are secured within balloon-expandable stents to facilitate deployment. The strengths and weaknesses of these valves are beyond the scope of this review; however, the selection largely depends on the patient’s anatomy [19,29]. Though effective at relieving PV insufficiency, bioprosthetic transcatheter valves (both surgical and transcatheter) have long-term complications related to somatic overgrowth, degeneration, calcification, rejection, the need for long-term anticoagulation and may lead to eventual replacement [18,30,31,32].

Tissue engineering offers a promising alternative as it has the potential to mitigate these complications due to the ability to create living, native tissue that grows with the patient [33]. Initial work using tissue-engineering strategies for PVR focused on decellularized xeno- and homografts seeded with various cell populations [34,35]. Despite relatively encouraging results, lengthy incubation periods required for in vitro cell culture rendered these tissue-engineered pulmonary valves (TEPVs) less than ideal, and has pushed groups to develop constructs without the need of cell seeding prior to implantation [36,37]. Further work in this field has sought to define an ideal PVR, with the goal of creating an “off-the-shelf” TEPV capable of self-renewal and growth within pediatric patients. While the concept of tissue engineering encompasses advantages over current technology [38], this section will focus solely on select studies regarding synthetic, bioabsorbable TEPV scaffolds.

The benefits of creating cardiac valves using tissue-engineering principles and synthetic, bioabsorbable biomaterials are remarkable [39]. Using synthetic, bioabsorbable materials alleviates the issue of limited material supply that many non-synthetic TEPVs constructs encounter. Additionally, as the body degrades the scaffolds and replaces it with its own tissue, the neo-valve has the potential to grow with the patient. This bypasses the need for additional replacement procedures secondary to PV insufficiency from somatic overgrowth. Synthetic materials also lend themselves to being more customizable as medicine continues to search for patient-specific solutions (Figure 1). Nonetheless, challenges remain in constructing a synthetic, fully resorbable TEPV limiting its widespread clinical use.

Just as current treatment options are delivered via transcatheter or surgical approaches, TEPVs have also been studied through both delivery modalities. In the transcatheter approach, the leaflets are completely degradable; however, current metal stents that house the valve are permanent. One manufacturing approach involves creating the structure ex vivo using autologous cells seeded onto a scaffold, culturing these constructs in a bioreactor to allow for ECM production, followed by decellularization and subsequent implantation. An example of this is the polyglycolic acid (PGA) and poly(4-hydroxybutyrate) PV that has undergone several changes to its design with positive large animal model results [43,44,45]. Another strategy avoided ex vivo culture altogether, using the TEPV recipient’s own immune system, along with a bioresorbable polymer bisurea-modified poly-carbonate scaffold, to assist with remodeling [46,47]. Although these valve designs have seen some exciting preclinical success, the long-term efficacy remains to be seen as animal models developed pulmonary regurgitation at earlier timepoints secondary to inappropriate valve regeneration, calcification or contracture. This is likely due to the complex structure of the native PV and the need for an equally complex TEPV in order to competently assume its role in controlling blood flow.

Regenerating such a structure involves an understanding of the delicate balance between cellular and molecular mechanisms as well as mechanical forces driving the regenerative process. Further studies aimed at elucidating the mechanisms driving the tissue regeneration process are needed before more effective clinical translation of TEPVs can occur. A mouse model of PV transplantation may assist in evaluating these mechanisms through the power of transgenic mice [48], albeit downsizing such mechanically complex structures may prove to be a separate but equally challenging hurdle. Furthermore, the role of long-term stent placement in this pediatric population is unknown. Studies evaluating stent placement in pulmonary stenosis not requiring PVR have complication rates ranging from 10 to 33%, and require frequent reinterventions [49]. The risk of damage to the valve in the event of a stent complication or to the stent in the case of a valvular intervention has yet to be fully described. Still in its beginning stages, a zinc–aluminum alloy degradable metal stent housing a synthetic, electrospun polycaprolactone valve is being used in a fetal transcatheter ovine model and may provide an alternative solution to the issue of long-term stent placement [42]. Despite zinc being an essential metal for many cellular processes, there are still only limited in vitro and in vivo data demonstrating its impact within the cardiovascular system and tissue regeneration pathway [50,51].

Although transcatheter approaches are an arguably more clinically attractive approach due to the decreased morbidity for the patient, surgically implanted synthetic TEPVs have seen the most success to date. Perhaps the most successful surgically implanted TEPV is that from Xeltis (Zurich, Switzerland). Made from a ureido-pyrimidinone supramolecular polymer, this valve-conduit structure has seen promising preclinical results with adequate hemodynamic profiles in sheep up to two years after implantation [52]. Currently, clinical trials are underway evaluating two different types of this TEPV in the Xplore-1 and Xplore-2 trials. Data presented at the International Conference of Tissue Engineered Heart Valves 2020 revealed that 11 out of 12 patients had severe pulmonary regurgitation at two years in the Xplore-1 trial. In the Xplore-2 trial, there were no signs of severe regurgitation; however, out of the 6 enrolled patients, 1 developed valve stenosis and 1 required reoperation at one year [53]. A similar issue is raised over the surgically placed biodegradable scaffolds, as severe pulmonary regurgitation would require reintervention and possibly additional PV replacement. Further work targeting the mechanisms driving regeneration would again be ideal to help guide additional modifications to scaffold design.

Regardless of the delivery method, tissue regeneration relies on a complex cell-mediated remodeling process inherent in many bioengineering approaches. The major obstacle for TEPVs is leaflet retraction and valvular insufficiency. Computational modeling offers a potential solution for this as a powerful tool that allows for consideration of many different mechanical and biological factors affecting valve design and performance. Simon Hoerstrup’s group has validated this concept, demonstrating that computational modeling could improve their PGA-based TEPV scaffold design and guide tissue remodeling for improved long-term performance and in vivo imaging results in sheep up to one year [54]. Though the logistical and technical challenges that come with validation of a computational modeling approach are many, this type of modeling is important for advancing the field of TEPVs in the search for a viable, bioabsorbable pulmonary valve replacement capable of self-repair, growth and remodeling. Nonetheless, we await continued studies in the field and are excited for the future of synthetic, bioabsorbable TEPVs.

## 3. Tissue-Engineered Patch

Patches are essential for repair of CHD when repairing congenital defects or stenosis. They are commonly used for tissue repair and reconstruction in congenital heart surgeries such as pulmonary angioplasty, aortic arch reconstruction, right ventricular outflow tract patch repair, valvular augmentation, and repair of the coronary sinus [55]. A variety of materials are used for the patch depending on the location and desired handling characteristics. Such materials include: decellularized homograft, autologous pericardium (fresh or glutaraldehyde treated), Dacron, Hemashield^®^, expanded polytetrafluoroethylene (ePTFE), as well as RVOT conduits [56,57]. These materials are essentially non-growing, resulting in the surgeon performing cardiac reconstruction with the hope that there will be sufficient growth of nearby autologous tissue to avoid the need for reoperation [57]. The decision of which patch type to use is commonly based on the area to be repaired and pressures the material will face in those locations. For example, in areas under high pressure loads, patches made from artificial materials such as Dacron and Hemashield^®^ are often used. In this case, however, the patient needs to be closely monitored during the acute postoperative period as significant hemolysis can occur from the blood flow jet interacting with the exposed patch. Long-term issues are mainly degeneration and calcification of the patch. In particular, patches made of heterogeneous pericardium can calcify and cause stenosis. There are high hopes that bioabsorbable graft materials combined with autologous tissue regeneration will be developed into a patch material able to handle high pressure loads without degeneration [58,59].

Repair of aortic arch hypoplasia or aortic arch coarctation is one of the most common congenital heart disease procedures requiring reintervention [60]. Reconstruction of the aortic arch involves many different operative techniques including resection with end-to-end anastomosis of the stenosis, patch augmentation of the hypoplastic aortic arch, subclavian flap repair, and artificial vessel replacement [61,62]. Complications related to this procedure include restenosis, aortic aneurysm formation, aortic dissection, rupture, residual hypertension, early onset of atherosclerotic lesions (cerebrovascular disease and coronary artery disease), and infective endocarditis [63,64]. The most common of these being re-development of aortic stenosis or coarctation. Even after repair of the stenosis, life expectancy does not normalize. Patients treated surgically at an average age of 16 years have reported 10, 20 and 30 years’ survival rates of 91%, 84%, and 72%, respectively [65]. Early surgical treatment is believed to improve long-term prognosis [66,67,68]; though surgical treatment at an average age of 5 years results in 20-year and 40–50-year survival rates of 91% and 50%, respectively. It has been reported that 70% of remote deaths are due to cardiac complications [69]. The creation of tissue-engineered arterial patches with growth potential can potentially eliminate the need for reoperation and improve the quality of life and survival rates in these patients.

Based on a decellularized porcine small intestinal mucosa, the CorMatrix^®^ patch technology demonstrates tissue remodeling and regenerative potential in preclinical experiments [70,71,72]. However, it fails to remodel into native-like tissue at 9 months after implantation in patients with congenital heart failure [73]. These non-native structures, under the intense physiological loads of the cardiovascular system, are limited in their long-term effectiveness. The development of a viable biodegradable material that has the mechanical strength to withstand fluctuating pressures during degradation and remodeling is central to the successful fabrication of a tissue-engineered vascular patch. Development of such a biodegradable material could pave the way for many other tissue-engineered bioproducts useful in cardiac surgery and would likely spur the development of various pre-clinical and clinical trials [11]. One group developed a tissue-engineered vascular patch from decellularized extracellular matrix (ECM) obtained during surgical removal of the human aorta and seeding it with bone marrow stem cells from pediatric patients [74,75]. This material was validated in canine abdominal aorta and showed excellent pressure resistance. However, even though it demonstrated formation of new endothelium and smooth muscle, it has yet to be determined if this patch material has significant growth potential.

Another group utilized a synthetic polymer approach to produce a hybrid biodegradable polymer scaffold from polylactide-co-epsilon-caprolactone (PLCL) copolymer reinforced with PGA fibers. Additionally, they took advantage of in vitro cell programing by inducing canine bone marrow mononuclear cells to differentiate into either vascular or endothelial cells, seeding them onto the scaffolds and subsequent implantation into the inferior vena cava of bone marrow donor dogs to create tissue-engineered vascular patches 15 mm wide × 30 mm long [76,77]. This study demonstrated that, compared to PLCL scaffolds, PGA/PLCL scaffolds exhibited tensile mechanical properties close to those exhibited by native canine inferior vena cava vessels. Furthermore, the investigators also confirmed that hybrid bioabsorbable polymer scaffold implanted bone marrow mononuclear cells survived after implantation and contributed to regeneration of endothelium and vascular smooth muscle in the implanted vascular patches.

A group led by Ichihara developed a novel fully biodegradable polymeric scaffold consisting of epsilon-caprolactone and lactide acid copolymer [P(CL/LA)] and poly-L-lactide acid (PLLA) that was transplanted as a 30 × 15 mm oval patch into the descending aorta of 12 dogs without any cell seeding (Figure 2) [78]. Cell proliferation was evaluated by histological and immunohistochemical methods (Figure 2D), and there was no macroscopic evidence of patch rupture or aneurysm formation up to 6 months (Figure 2A–C). This suggested that their bioabsorbable polymeric scaffold could be used as an alternative vascular material for high-pressure systems, and since the material is completely degradable, it may also have the desirable potential to grow. While only a few studies focused specifically on larger arterial patches, there may be helpful clues to be gained from the study of small diameter arterial grafts that can be translated to large patch materials in similar high-pressure arterial environments [11].

## 4. Regenerative Medicine Solutions for Pulmonary Vein Stenosis

Pulmonary vein stenosis (PVS) can either be congenital or a postoperative complication of total anomalous pulmonary venous connection (TAPVC) repair (9-18%) and has a poor prognosis (Figure 3) [79,80]. The most severe form of PVS is pulmonary vein obstruction. Though the exact mechanism of PVS remains unclear, the histopathological changes related to development of PVS include intimal thickening at the surgical anastomosis sites and extension of the anastomosis into the pulmonary veins of the lung parenchyma [81]. Surgery is considered the preferred approach in most cases of congenital or acquired PVS with severe symptoms [82]. Specifically, (1) endarterectomy (resection of the stenotic ring and anastomosis of the pulmonary vein directly to the left atrial endocardium) and (2) pericardial patch venoplasty (resection of the stenotic tissue and anastomosis of the patch to enlarge the stenotic segment) are the most common traditional techniques (Figure 3A). Newer, suture-less marsupialization techniques providing direct adhesion of the pericardium surrounding the affected pulmonary vein to the left atrium (thus avoiding direct suturing of the cut end of the vessel) have been reported to help reduce the risk of restenosis by preventing suture line deformation and reducing tissue growth stimulation [80]. Overall, published surgical outcomes are moderate, with only half of cases being free of restenosis or death after several years [83,84]. Furthermore, pneumonectomy may be indicated in cases of severe or uncontrolled hemoptysis, and lung transplantation has been performed in patients with severe pulmonary hypertension resulting from PVS [85].

Another treatment modality for PVS is transcatheter balloon angioplasty and stent implantation (Figure 3B). This approach is challenging as it can require high-pressure balloons, and stents which can be eventually expanded to an adult size of >12 mm as the patient grows. Results from these types of interventions are suboptimal, requiring frequent re-dilation due to high rates of restenosis [86]. Early study of bare metal stents (BMS) in patients with PVS and CHD showed that freedom from occlusion or severe in-stent stenosis at 1 year was 37%. In an effort to alleviate this complication, drug-eluting stents (DES) have been increasingly utilized for the treatment of PVS [87]. In addition, DES have been shown to be more effective in treating PVS than BMS. Stented pulmonary veins are at risk for somatic overgrowth requiring reintervention which has led to further studies to avoid this process [88].

Several groups are conducting translational research to solve the mechanism of PVS in an effort to find new solutions using known drugs for this challenging disease process (Figure 3C). Saiki et al. have created an animal model of postoperative PVS in pigs and conducted several studies to elucidate the mechanism of intimal thickening in the pulmonary veins of the lung parenchyma [89]. They confirmed the proliferation of secretory smooth muscle-like cells as the main change in the pulmonary veins within the lung parenchyma. Activation of the mTOR pathway was observed in the pulmonary veins within the lung parenchyma, suggesting that this pathway may also contribute to proliferation. To further investigate this, extravascular application of a sustained-release film of rapamycin (an immunosuppressive agent) to the anastomosis demonstrated inhibition of intimal thickening at the anastomosis and mTOR activity. This suggests a possible new therapeutic strategy for PVS prevention. Zhu et al. have previously reported a piglet model of PVS with progressive diffuse obstructive intimal hyperplasia in the upstream pulmonary veins, recapitulating the clinical pathogenesis of PVS [90,91]. Specimens from the upstream zonal pulmonary veins of the pig model and human PVS patients were associated with robust expression of transforming growth factor-β (TGF-β). Systemic administration of losartan, a known TGF-β inhibitor, ameliorated the pulmonary hypertension and intimal hyperplasia associated with PVS. This indicates the medication’s potential usefulness as a prophylactic treatment for patients at high risk of developing PVS after pulmonary vein surgery. Similarly, Rehman et al. reported a study of vinblastine and methotrexate as therapies for infants and children with progressive multivessel PVS, targeting the presence of myo-fibroblasts within the lesion [92]. Quinonenz et al. also mentioned the effectiveness of chemotherapeutic agents that target neointimal proliferation being used to treat PVS [93,94]. A prospective, open-label clinical study of the use of bevacizumab and imatinib has shown an overall improvement in disease progression and patient survival in patients with multivessel PVS [95]. Taken together, these findings suggest that the prognosis of pulmonary vein stenosis may be improved by controlling excessive intimal thickening associated with post-anastomotic inflammation and rapid somatic growth in children.

Replacing materials such as conventional stents, patches, and suture with bioabsorbable materials would be very beneficial in preventing PVS. For example, the current generation of bioabsorbable drug-eluting platforms have been evaluated in coronary artery disease and have shown strong results. In the future, there may be a crossover to the treatment of pediatric PVS. Bioabsorbable stents are designed to support the body conduit only during its healing process, the mass and strength of the stent decreases with time, and the mechanical load is gradually transferred to the surrounding tissue. The bioabsorbable stent also allows for longer term delivery of drugs from the internal reservoir to the conduit wall, eliminating the need for a second surgery to remove the device [96]. The development of bioabsorbable suture may also inhibit intimal thickening of the anastomosis, similar to stents. Padmakumar has developed a biodegradable, drug-releasing suture that prevents intimal thickening [97].

Since bioabsorbable materials such as patches are associated with excessive inflammation and material thickening during the process of new tissue formation [85], the inhibition of intimal thickening by the addition of biological inhibitors may also play a significant role in resolving this phenomenon. We have created a vascular patch with sustained release of rapamycin and are currently testing its efficacy in a sheep model. The patch is still the first choice for neonatal TAPVC repair, and we hope that the development of this research will change the prognosis of pediatric patients.

## 5. Tissue-Engineered Vascular Graft

Modified Fontan surgery for the treatment of single ventricle anomalies is still palliative and not curative [98]. Commonly cited complications associated with the use of synthetic conduits in extracardiac total cavopulmonary connection include thromboembolic events and stenosis development [3,4,99,100]. Traditionally, surgeons have used various materials and techniques such as Dacron, ePTFE, homografts, autologous pericardium, and heterologous pericardium for extracardiac Fontan surgery. Many studies have reported strong mid-term results up to 10 years using some of these extracardiac Fontan conduits [101,102]. However, since there is no long-term follow-up of these materials nor precise data on how many patients end up with revision surgery, etc., there are high expectations for tissue-engineered grafts to fill the void left from the previously listed materials.

Tissue-engineered vascular grafts (TEVGs) able to mimic native tissue and grow with a patient have been highly studied for more than a decade now and have been extensively reviewed over that period as well. Figure 4 highlights the growth potential of TEVGs as it presents unpublished data from the original Shinoka et al., 2005 study that shows the same patient’s pulmonary artery, graft and IVC measurements at 1 year (Figure 4A), 8 years (Figure 4B), and 11 years old (Figure 4C) [103]. More recently, Bockeria et al. in Switzerland has provided human data on their clinical experience with a new bioresorbable vascular graft based on electrospun polycaprolactone (PCL) in the Fontan circulation [104]. Interestingly, histology at 1-year post-implant from an ovine model showed that a significant amount of polymeric material still remained in the vascular tissue, leaving the less-than-ideal possibility that a long-term foreign body reaction (similar to that of Gore-Tex) from a slowly degrading material may occur.

In collaboration with a Japanese textile company, our group has created a scaffold by inserting a sponge layer made of PLCL into the PGA skeleton, producing a bioabsorbable and biocompatible scaffold. Early clinical trials in human patients have shown no aneurysm formation, graft rupture, graft infection, or calcification. Of note, 7 out of 25 patients developed stenosis and underwent at least one balloon angioplasty, with none of the patients requiring surgical revision or replacement of the graft [105,106]. Subsequently, our design received FDA approval for implantation in four patients in the United States with each showing gradual growth of the conduit without severe complications such as death. Despite the clinical success, we again observed a high rate of graft stenosis in the first 6 months. To elucidate the mechanism of this early stenosis, we developed a computational model for our TEVG using a data-driven design (Figure 5). Our model predicted the early stenosis as observed in clinical trials, but surprisingly it suggested that the stenosis would spontaneously reverse though an inflammatory-driven, mechano-mediated mechanism [107]. Further validation of this model occurred through implanting TEVGs in a sheep inferior vena cava interposition graft model, which confirmed the prediction that TEVG stenosis would resolve spontaneously [108]. These results suggested that although the need for appropriate medical monitoring remains important, angioplasty could have been safely avoided in patients with asymptomatic initial stenosis. Going one step further, the simulations also predicted that the degree of reversible stenosis could be mitigated through modification of the scaffold design to attenuate early inflammation, suggesting a new paradigm for optimizing the next generation of TEVGs. Based on these results, a new and improved second-generation graft has been created and FDA approval obtained for a new single institute Phase 2 clinical trial.

Despite the increasing success of this TEVG, there are still a few problems that have yet to be addressed. In particular, there is the issue of the Fontan procedure itself and its elimination of hypoxemia via the creation of a passive blood flow from the systemic venous system to the pulmonary circulation. Patients live in a state of preload deprived cardiac output due to these hemodynamic changes, resulting in increased systemic venous pressure. Increased venous pressure can lead to complications such as heart failure, arrhythmias, protein losing gastroenteropathy, pulmonary arteriovenous fistulas, and cirrhosis [103]. These complications ultimately contribute to increased morbidity and reduced long-term survival. Next-generation technology using “smart materials” that respond to physical or chemical stimuli may prove essential in compensating for these abnormal hemodynamics by creating grafts that can pulse or beat to assist with musculature and/or vascular contraction. Through the attachment of cardiomyocyte-induced pluripotent stem cells to the graft [109], or via liquid crystalline elastomers that create movement in response to a chemical stimulus [110], contractile grafts able to contribute to the hemodynamic improvement of Fontan patients are expected to appear in the future.

## 6. Conclusions

This review offers a unique and up-to-date interpretation of the real regenerative medicine needs in the treatment of congenital heart disease by highlighting important studies within 4 domains from a pediatric cardiovascular surgery perspective. Indeed, there have been many advancements towards application of tissue engineering in the field of pediatric cardiovascular surgery, thus it is important for future research studies to continue to demonstrate the efficacy and safety of these materials prior to human trials. This commonly occurs through the utilization of small and large animal models; however, these studies must do their best to utilize the 3 Rs of animal use (reduction, refinement, and replacement) in order to protect the animals. In studying these materials, the goal from a pediatric cardiovascular surgery perspective is to offer growth potential while being resistant to the development of thrombosis, stenosis, calcification, and infection. At the same time, the structure should have minimal to no effect on the cardiovascular system’s hemodynamics. Although further mechanistic and preclinical work is required, there is great promise in the use of bioabsorbable, synthetic materials as conduits, vascular patches, pulmonary artery valves, and in the treatment of pulmonary vein stenosis.

## Figures and Tables

**Figure 1 biomedicines-09-00478-f001:**
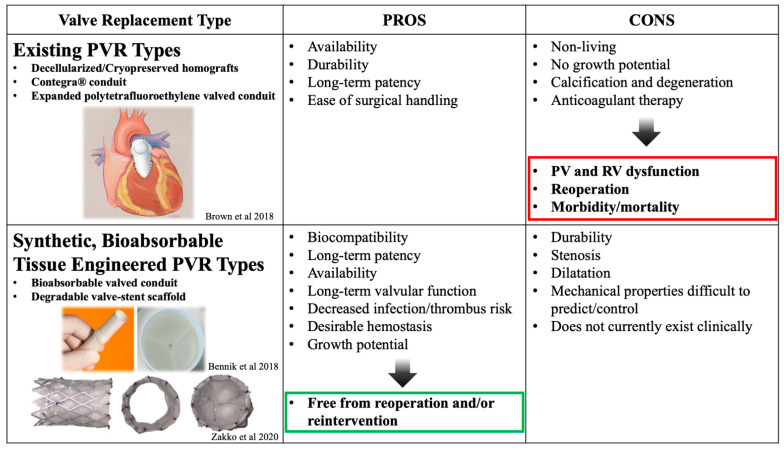
Pros and cons of current existing PVR options and a theoretical TEPV for PVR. The image in the top left is from an editorial on a PTFE RV-PA conduit [40]. The images in the bottom left box are of different views of the Xeltis biodegradable pulmonary valve conduit [41] from a bioabsorbable metal stent PVR from Zakko et al., 2020 [42]. PVR: Pulmonary valve replacement; PV: pulmonary valve; RV: Right ventricle; the green and red squares are to emphasize.

**Figure 2 biomedicines-09-00478-f002:**
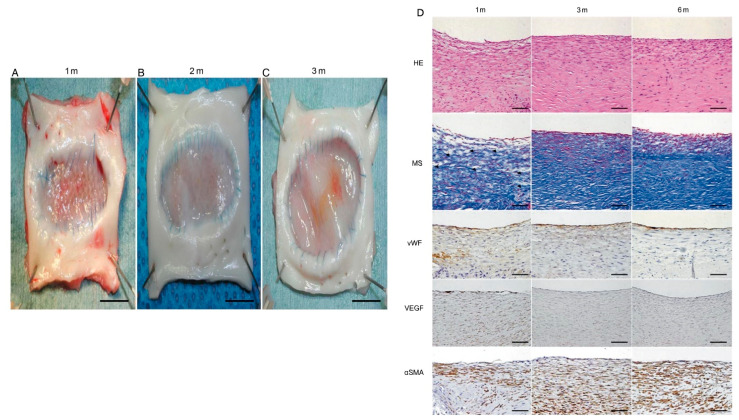
Data originally from Ichihara et al., 2015 demonstrating the macroscopic and histologic evaluation of a tissue-engineered arterial patch (TEAP) [78]. Macroscopic findings of each explanted tissue-engineered arterial patch at 1 month (**A**), 3 months (**B**) and 6 months (**C**) after implantation, showing no intimal hyperplasia and no aneurysm formation. No thrombus was observed. All inner sides of the patch were covered with native intima-like tissue with a smooth surface. Scale bar, 10 mm. (**D**) Hematoxylin–eosin (HE) and Masson staining (MS) at each observation end-point of bioabsorbable aortic patch. White and blank spaces (residual sponge polymer, arrowhead), which were observed in the TEAP at 1 month, were occupied by cellular components (red) and extracellular matrices (blue) in MS at 3 and 6 months after implantation. These findings showed good cell and tissue proliferation in the TEAP over time. Immunohistochemical staining for vWF, VEGF and αSMA (lower) is also shown. The luminal surfaces of the TEAPs were covered with a single layer of endothelial cells stained with antibodies to vWF at 1 month. The vWF-positive cells accumulated more clearly on the luminal surfaces of TEAP at 6 months. VEGF-positive cells in the patches were observed more on both the luminal surfaces and within the media 1 month after implantation; however, no VEGF expression was detected at 3 and 6 months. αSMA-positive cells were observed in the media of the regenerated tissue, and gradually increased over time. Original magnification HE, MS, vWF and αSMA ×200, VEGF ×100; scale bar, 100 μm. HE: haematoxylin–eosin; MS: Masson staining; TEAPs: tissue-engineered arterial patches; vWF: von Willebrand factor; VEGF: vascular endothelial growth factor; αSMA: alpha-smooth muscle actin.

**Figure 3 biomedicines-09-00478-f003:**
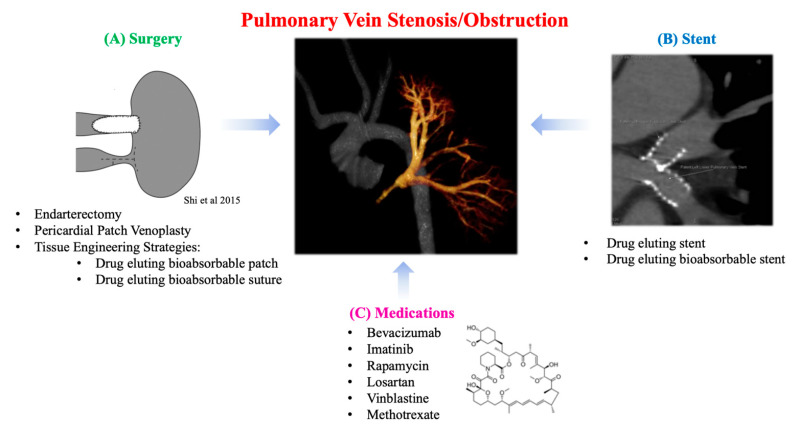
Treatment modalities for pulmonary vein stenosis/obstruction. The middle image is a clinical CT reconstruction from a pediatric patient who had pulmonary vein obstruction. (**A**) Describes surgical treatment options for pulmonary vein stenosis/obstruction and lists two potential tissue-engineering strategies for surgical management. Image is from Shi et al., 2015 [83]. (**B**) Clinical CT image from the patient who underwent stent placements for treatment of pulmonary vein obstruction. (**C**) Medications that have been studied to help in the treatment of pulmonary vein stenosis/obstruction. In 2018, the ethics committee at Nationwide Children’s Hospital approved the percutaneous intervention of pulmonary vein obstruction in human subjects. Informed consent was obtained from each patient, or from the parent/guardian if the patient was a minor, before proceeding.

**Figure 4 biomedicines-09-00478-f004:**
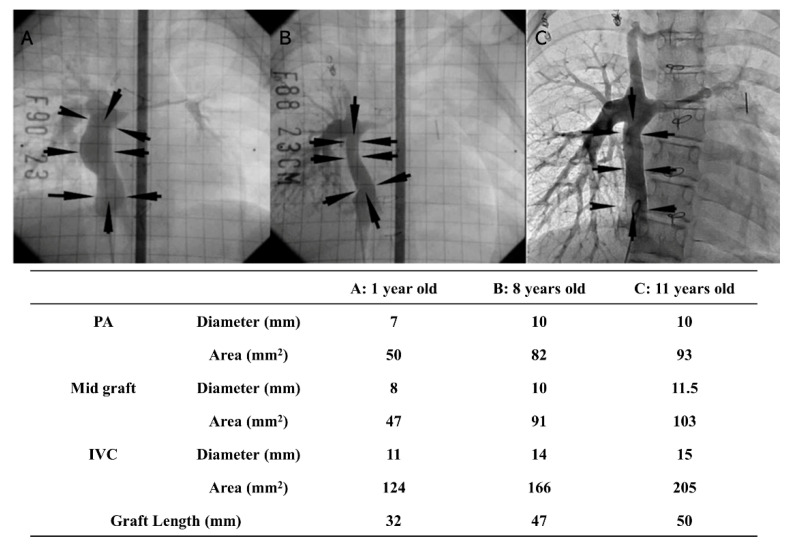
Three angiographic images in the same patient at different ages demonstrating growth of TEVG. (**A**) 1 year old, (**B**) 8 years old, (**C**) 11 years old. Arrows denote extracardiac TCPC graft. PA: Pulmonary artery; IVC: inferior vena cava. In 2001, the ethics committee at Tokyo Women’s Medical University approved the implantation of TEVGs in human subjects (IRB#198). The following inclusion criteria were used for patient screening elective surgery: age younger than 30 years, full understanding of the procedure by the patient or family, and minimal extracardiac disease burden. Informed consent was obtained from each patient, or from the parent/guardian if the patient was a minor, before proceeding.

**Figure 5 biomedicines-09-00478-f005:**
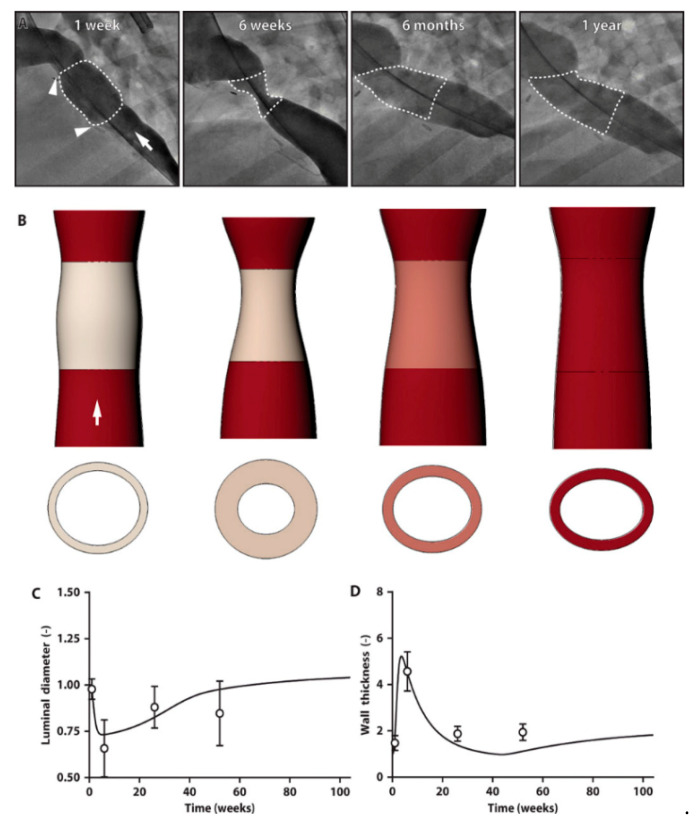
TEVG stenosis spontaneously reverses in an ovine model [108]. (**A**) Serial angiographic images of a representative ovine TEVG 1 week, 6 weeks, 6 months, and 1 year after implantation. Anastomoses are identified with white arrowheads, and the TEVG is identified by a white dotted outline. White arrow indicates direction of blood flow. (**B**) Image reconstructions of serial IVUS and angiographic measurements of luminal area, wall thickness, and length of the TEVG, confirming the development of stenosis at 6 weeks but spontaneous resolution by 6 months after implantation. White arrow indicates direction of blood flow. When based on lamb data, the model simulations (solid lines) for the midgraft. (**C**) Luminal diameter and (**D**) wall thickness fit well the hydraulic diameters calculated from the IVUS measurements (symbols) over the 1-year study. Ovine IVUS measurements are represented as means ± SD (*n* = 22 at 1 week, *n* = 22 at 6 weeks, *n* = 20 at 6 months, *n* = 15 at 1 year). The Institutional Animal Care and Use Committee of Nationwide Children’s Hospital (Columbus, OH, USA) reviewed and approved the protocol for the ovine study (AR13- 00079 approval date: 7 October 2020). Representatives of the animal care staff monitored all animals intraoperatively and during their postoperative courses. Animal care was within the humane guidelines published by the Public Health Service, National Institutes of Health (Bethesda, MD) in the care and use of laboratory animals (2011), as well as within U.S. Department of Agriculture regulations set forth in the Animal Welfare Act (Washington, DC, USA).

**Table 1 biomedicines-09-00478-t001:** CHD requiring treatment with RVOT valved conduit or pulmonary valve [22]. CHD: Congenital heart disease; RVOT: Right ventricular outflow tract; VSD: ventricular septal defect; TGA: transposition of the great arteries; DORV: double outlet right ventricle.

Classification of Diseases		Examples of Diagnoses
RVOT Congenital Defect	Stenosis	Tetralogy of Fallot +/− absent pulmonary valve syndrome
		Isolated pulmonary stenosis
	Atresia	Pulmonary atresia +/− VSD
		Truncus Arteriosus
RVOT Iatrogenic Defect	Rastelli Procedure	DORV + VSD + sub-pulmonary stenosis
		TGA or Corrected TGA + sub-pulmonary stenosis
	Ross Procedure	Aortic Stenosis (congenital or acquired)
		Aortic Regurgitation (congenital or acquired)
	Secondary Pulmonary Regurgitation	PR after repair of Tetralogy of Fallot
		PR after pulmonary valvuloplasty

## Data Availability

Data supporting reported results of this study can be found in publiclyarchived datasets as specified in links throughout the manuscript.

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
