# Peer review of "The Real Need for Regenerative Medicine in the Future of Congenital Heart Disease Treatment"

_biomedicines, 2021, doi:10.3390/biomedicines9050478_

Round 1
Reviewer 1 Report
Dear authors,
The authors wrote a review about the treatment of congenital heart disease in the field of regenerative medicine, especially in the pediatric cardiovascular surgery field. They summarize the availability of bioabsorbable materials that can be used to replace autologous tissues, because they can replace (in term of growth potential) materials that are used but these last ones have limitations such as uncontrolled growth of the grafts. Very few materials are used in clinical trial because there is a gap between the material developed in laboratories and the ones that are needed by the surgeons. The authors focused on different aspect of cardiovascular surgeries such as tissue-engineered pulmonary valve, patch, pulmonary vein stenosis and vascular graft. The manuscript is well written, and the authors present well the pros and cons of the different bioabsorbable materials for the different cardiovascular surgeries. They finally present a new scaffold, approved by the FDA, used in clinical trials that could be used in vascular graft.
I have some comments about the manuscript:
Major:
- Figure 3: Are the pictures obtained from a patient or animals? It should be mentioned in the legend
- Figure 3/4: Based on the journal policy (https://www.mdpi.com/journal/data/instructions), and if the data are from a human study, “When reporting on research that involves human subjects, human material, human tissues, or human data, authors must declare that the investigations were carried out following the rules of the Declaration of Helsinki of 1975 (https://www.wma.net/what-we-do/medical-ethics/declaration-of-helsinki/), revised in 2013. According to point 23 of this declaration, an approval from an ethics committee should have been obtained before undertaking the research. At a minimum, a statement including the project identification code, date of approval, and name of the ethics committee or institutional review board should be stated in Section ‘Institutional Review Board Statement’ of the article. Data relating to individual participants must be described in detail, but private information identifying participants need not be included unless the identifiable materials are of relevance to the research (for example, photographs of participants’ faces that show a particular symptom). Editors reserve the right to reject any submission that does not meet these requirements.”. IF the data are from other publications, the authors should obtain the permission for copyrights.
- Figure 5: the animal study Number and the approval by an IACUC committee should be provided.
Minor:
- All latin words must be written in Italic (In vivo, vena cava…).
- In the legend of figure 2, A/B/C should be mentioned, and the size of A/B/C scale bar should be mentioned in the legend.
- Line 396: Ovine should be written ovine.
Author Response
REVIEWER 1
Major:
Figure 3: Are the pictures obtained from a patient or animals? It should be mentioned in the legend
- Pg 8. These are both previously unpublished clinical images from a patient that suffered from pulmonary vein obstruction that were obtained after acquiring the appropriate patient or parent/guardian consent. This has been addressed in the legend Pg 8 lines 325-329.
Figure 3/4: Based on the journal policy (https://www.mdpi.com/journal/data/instructions), and if the data are from a human study, “When reporting on research that involves human subjects, human material, human tissues, or human data, authors must declare that the investigations were carried out following the rules of the Declaration of Helsinki of 1975 (https://www.wma.net/what-we-do/medical-ethics/declaration-of-helsinki/), revised in 2013. According to point 23 of this declaration, an approval from an ethics committee should have been obtained before undertaking the research. At a minimum, a statement including the project identification code, date of approval, and name of the ethics committee or institutional review board should be stated in Section ‘Institutional Review Board Statement’ of the article. Data relating to individual participants must be described in detail, but private information identifying participants need not be included unless the identifiable materials are of relevance to the research (for example, photographs of participants’ faces that show a particular symptom). Editors reserve the right to reject any submission that does not meet these requirements.” IF the data are from other publications, the authors should obtain the permission for copyrights.
- Thank you bringing this to our attention. We have added clarification statements regarding the consent or IRB approvals for the studies in figures at Pg 8 lines 330-332 and at Pg 11 Lines 419-423.
Figure 5: the animal study Number and the approval by an IACUC committee should be provided.
- Pg 12 line 460-466 the IACUC protocol number and approval statement has been added.
Minor:
All latin words must be written in Italic (In vivo, vena cava…).
- In vivo, in vitro and vena cava have been italicized throughout the document.
In the legend of figure 2, A/B/C should be mentioned, and the size of A/B/C scale bar should be mentioned in the legend.
- Pg 7 line 287 clarification of A, B, C has been added.
- Pg 7 line 289 scale bar added.
- Pg 8 line 300 scale bar added.
Line 396: Ovine should be written ovine.
- Pg 10 Line 412 “Ovine” has been changed to “ovine”
Reviewer 2 Report
The review touches on many biomaterials that have reported to date as promising treatments for congenital heart diseases. The analysis is well performed and of interest for people working in the material or bioengineering field. This work particularly focuses on bioabsorbable materials. The possible adversity of non-absorbable materials that failed in the preclinical or clinical studies are here reported. I would like to have few comments, if authors know, about the possibility of implanting the so-called smart materials, i.e., responding to physical or chemical stimuli used to assist muscle or vasculature contraction.
For instance, light-activated elastomers have been recently suggested as a biocompatible material for the design of actuators assisting contraction (doi: 10.1161/CIRCRESAHA.118.313889)
Author Response
Reviewer 2
The review touches on many biomaterials that have reported to date as promising treatments for congenital heart diseases. The analysis is well performed and of interest for people working in the material or bioengineering field. This work particularly focuses on bioabsorbable materials. The possible adversity of non-absorbable materials that failed in the preclinical or clinical studies are here reported. I would like to have few comments, if authors know, about the possibility of implanting the so-called smart materials, i.e., responding to physical or chemical stimuli used to assist muscle or vasculature contraction.
For instance, light-activated elastomers have been recently suggested as a biocompatible material for the design of actuators assisting contraction (doi: 10.1161/CIRCRESAHA.118.313889)
- Pg 12 Lines 468-482 Thank you for this suggestion. We have included a paragraph in the TEVG section highlighting the usefulness of these smart materials for improving hemodynamic forces in Fontan circulation. We also have added two references related to this References 108 and 109.
- Shi, X.; He, L.; Zhang, S.M.; Luo, J. Human iPS Cell-derived Tissue Engineered Vascular Graft: Recent Advances and Future Directions. Stem Cell Rev. Reports 2020, doi:10.1007/s12015-020-10091-w.
- Ferrantini, C.; Pioner, J.M.; Martella, D.; Coppini, R.; Piroddi, N.; Paoli, P.; Calamai, M.; Pavone, F.S.; Wiersma, D.S.; Tesi, C.; et al. Development of Light-Responsive Liquid Crystalline Elastomers to Assist Cardiac Contraction. Circ. Res. 2019, 124, e44–e54, doi:10.1161/CIRCRESAHA.118.313889.
Reviewer 3 Report
The paper entitled ‘What is the real need for regenerative medicine in the future of congenital heart disease treatment?’ deals with a very interesting topic in which materials science and regenerative medicine are giving exceptional results.
Given the great experience of the authors and the large number of articles published in this field, I would propose to them major revision to enhance the quality of their manuscript. I would like to try to give them some advice on how to make the text and figures clearer and more comprehensive.
The most important thing is that the authors make the colleagues who read them understand what the real purpose of their review is, by making the information they provided in the various sections more coherent. In particular, the abstract and the conclusions must be rewritten after having restructured the main text. The main text and table must be written in such a way that the state of the art of congenital heart disease treatment is clearly distinguished from the proposals of the authors or the latest news in the field of research in materials science and regenerative medicine and biotechnology.
The authors are also required to indicate the origin of all panels of the figures they present if published in previous works, moreover each panel should be cited and described in the main text. The quality of the figures can certainly be improved by increasing the resolution of the texts inserted in them and eliminating some typo. In any case, to help the authors in the revision of their paper which I hope will be published soon, I have included in the pdf text a series of comments that I hope will be useful.

Author Response
Title
I personally believe that the title used by the authors is more suitable for an opinion or an editorial and not for a review. I wuold kindly ask authors to rethink the title of the review avoiding the interrogative form and adding the reference to the pediatric cardiovascular surgery.
- Pg 1 lines 2-3: We have adjusted the title to remove the interrogative nature of it. The new title is: “The real need for regenerative medicine in the future of congenital heart disease treatment”.
Abstract
The abstract is too generic and makes no reference to pediatric cardiovascular surgery. In the abstrac in the keywords authors reference i to large animal models but in the main text this is not adequately described.
specify the 4 areas
- Pg 1 lines 19-28: We appreciate the concerns and have adjusted the abstract to make it less generic by specifying the four areas of tissue engineered pulmonary valves, tissue engineered patches, regenerative medicine options for treatment of pulmonary vein stenosis and tissue engineered vascular grafts. We added the phrase “pediatric cardiovascular surgery” twice. We removed the “large animal model” statement.
Introduction
[Fabrication and Applications of Micro/Nanostructured Devices for Tissue Engineering. Nano-Micro Lett. 9, 1 (2017). https://doi.org/10.1007/s40820-016-0103-7]]
[Three-dimensionally two-photon lithography realized vascular grafts. Biomed Mater. 2020 Nov 13. doi: 10.1088/1748-605X/abca4b. Epub ahead of print. PMID: 33186926.]
- Pg 2 Line 49 references 6 and 7 These references have been added.
Dzilic E., Doppler S., Lange R., Krane M. (2019) Regenerative Medicine for the Treatment of Congenital Heart Disease. In: Serpooshan V., Wu S. (eds) Cardiovascular Regenerative Medicine. Springer, Cham. https://doi.org/10.1007/978-3-030-20047-3_11
Ambastha C, Bittle GJ, Morales D, Parchment N, Saha P, Mishra R, Sharma S, Vasilenko A, Gunasekaran M, Al-Suqi MT, Li D, Yang P, Kaushal S. Regenerative medicine therapy for single ventricle congenital heart disease. Transl Pediatr. 2018 Apr;7(2):176-187. doi: 10.21037/tp.2018.04.01. PMID: 29770299; PMCID: PMC5938254.
- Pg 2 Line 61 references 12 and 13: These 2 references on regenerative medicine have been added.
Tissue Engineered Pulmonary Valve
Table 1
the table should be better formatted including refs
- Pg 3 Table 1 has been reformatted and uploaded as a high-resolution TIFF. The legend has also been updated with the appropriate reference Pg 3 line 91 reference 22:
- Yamamoto, Y.; Yamagishi, M.; Miyazaki, T. Current status of right ventricular outflow tract reconstruction: complete translation of a review article originally published in Kyobu Geka 2014;67:65–77. Thorac. Cardiovasc. Surg. 2015, 63, 131–141, doi:10.1007/s11748-014-0500-0
Table 2
the resolution is low and he table must be reformatted
- Pg 4 Table 2 has been reformatted and reuploaded as a high-resolution TIFF. Additionally, the images within the figure have been cited within the figure and in the legend of the figure. Pg 4 lines 140-142 references 40 and 41.
- Brown, J.W. Polytetrafluoroethylene valved conduits for right ventricle–pulmonary artery reconstruction: Do they outperform xenografts and allografts? Thorac. Cardiovasc. Surg. 2018, 155, 2577–2578, doi:10.1016/j.jtcvs.2018.01.019.
- Zakko, J.; Blum, K.M.; Drews, J.D.; Wu, Y.L.; Hatoum, H.; Russell, M.; Gooden, S.; Heitkemper, M.; Conroy, O.; Kelly, J.; et al. Development of Tissue Engineered Heart Valves for Percutaneous Transcatheter Delivery in a Fetal Ovine Model. JACC Basic to Transl. Sci. 2020, 5, 815–828, doi:10.1016/j.jacbts.2020.06.009.
Reviewer 3 Highlighted “from the same group”
- Pg4 line 152 “From the same group” has been removed from the sentence.
Tissue Engineered Patch
Figure 1
the figures have low resolution. In the text the references to the two panels 'a' and 'b' do not appear and references from which the figures were taken are not indicated in the figure or if unpublished by the authors
- Pg 6 Figure 1 Thank you for the comment. After further consideration of the manuscript we have decided to remove figure 1 completely. The manuscript has been edited to reflect this by adjusting the other figure numbers as well by removing reference to figure 1 on Pg 6 line 231.
****** FROM HERE ON THE FIGURES WILL BE REFERRED TO BY THEIR UPDATED FIGURE NUMBER****** (i.e. the previous figure 2 will now be considered figure 1)
New Figure 1
Fonts are small and figure should be reformatted. in the legend the references are missed as links to the different panels in the main text
- Pg 7 The figure has been reuploaded as a high-resolution TIFF. Additionally, the description of the figure in the legend has been updated to clarify A, B, C, D on Pg 7 lines 285- Pg 8 line 301
in the main text link ti Figure 2 is missed
- Pg 7 line 276 The link to figure 1 has been added.
- Pg 7 Lines 277 and 278-279 labeling of A, B, C, D panels in the main text.
Regenerative Medicine Solutions for Pulmonary Vein Stenosis
conform to the other paragraphs for example ... Pulmonary Vein Stenosis tissue engineering solution
- Pg 8 line 302 The heading of the section has been adjusted to “Regenerative medicine solutions for Pulmonary Vein Stenosis”.
New Figure 2
the figure have a low resolution, is not clear also for content. In the text there are no references to panels a and b, The legend is not explanatory and the red marks of the ppt assembly remain in the figure
- Pg 8 Figure 2: The figure has been reformatted, red marks removed and reuploaded in high-resolution with citation added Pg 8 line 328 reference 82
- Shi, G.; Zhu, Z.; Chen, H.; Zhang, H.; Zheng, J.; Liu, J. Surgical repair for primary pulmonary vein stenosis: Single-institution, midterm follow-up. Thorac. Cardiovasc. Surg. 2015, 150, 181–188, doi:10.1016/j.jtcvs.2015.03.032.
- Pg 8 lines 325-332 updated the legend description to identify A, B and C.
- To clarify the where A, B and C related to the main text, links have been added to the corresponding paragraphs Pg 8 line 313 Figure 2A, Pg 9 line 334 Figure 2B, Pg 9 line 345 Figure 2C.
Tissue Engineered Vascular Graft
New Figure 3
insert refs.in the main text should be insert to the figure panels
- Pg 10 Line 407 This was previously unpublished data from the original study performed by Shinoka et al 2001 “Midterm clinical result of tissue-engineered vascular autografts seeded with autologous bone marrow cells”. This has been mentioned in the main text.
- Pg 10 Lines 406-409 These images are of the same patient at 1 year (A) 8 years (B) and 11 years old (C). We have added clarification of this in the main text.
- Pg 11 Lines 420-423 added a statement on consent for the study and images in the figure legend.
New Figure 4
ref is missed
- Pg 12 line 442 reference 107 reference has been added
- Drews, J.D.; Pepper, V.K.; Best, C.A.; Szafron, J.M.; Cheatham, J.P.; Yates, A.R.; Hor, K.N.; Zbinden, J.C.; Chang, Y.C.; Mirhaidari, G.J.M.; et al. Spontaneous reversal of stenosis in tissue-engineered vascular grafts. Transl. Med. 2020, 12, 1–14, doi:10.1126/scitranslmed.aax6919.
Conclusions
the authors must highlight the information that this review wants to give with respect to the wide range of already existing references
- Pg 13 line 484-486 Thank you for this comment. As technology advances quickly, we feel that there was a need for a review like this that uniquely highlights these 4 domains from pediatric cardiovascular surgery perspective.We have added more to the conclusion that highlights that this specifically.
authors should better argue and explain this statement since according to EU legislation and several national and international initiatives such as 3R Centers and networks, it is time to develop advanced alternatives to animal testing and to encourage their application towards reduction, refinement, and even replacement (3R) of animal use for scientific purposes and in pre-clinical research.
- Pg 13 lines 489-492 This is important so we have added comments in the conclusion addressing the need for future studies to do their best to reduce, refine and replace animal models wherever possible in research.
Round 2
Reviewer 1 Report
Dear Authors,
I have no additional comments.
Sincerely
Author Response
Thank you so much !
Reviewer 3 Report
I really appreciate the review work colleagues have done on their manuscript, I would just like to recommend them to improve the quality of Table 2 and Figure 2
Author Response
The reviewers’ comments were well received and appreciated as we feel it has further strengthened our manuscript. We have addressed their concerns of our table and figure quality by adding additional surgery-specific language and pertinent images to the table as well as making minor adjustments to both that improve the overall quality. We also have included even higher resolution TIFFs to ensure that the issue of quality is not due to poor resolution.